# Toxicity of Asciminib in Real Clinical Practice: Analysis of Side Effects and Cross-Toxicity with Tyrosine Kinase Inhibitors

**DOI:** 10.3390/cancers15041045

**Published:** 2023-02-07

**Authors:** Lucía Pérez-Lamas, Alejandro Luna, Concepción Boque, Blanca Xicoy, Pilar Giraldo, Raúl Pérez López, Concepción Ruiz Nuño, Natalia De las Heras, Elvira Mora Casterá, Javier López Marín, Adrián Segura Díaz, Valle Gómez, Patricia Vélez Tenza, Magdalena Sierra Pacho, Juan Antonio Vera Goñi, Melania Moreno Vega, Alberto Alvarez-Larrán, Montse Cortés, Manuel Pérez Encinas, Patricia Carrascosa Mastell, Anna Angona, Ana Rosell, Sunil Lakhwani, Mercedes Colorado, Elena Ramila, Carlos Cervero, Beatriz Cuevas, Lucía Villalón Blanco, Raquel de Paz, Antonio Paz Coll, María José Fernández, Luis Felipe Casado, Juan Manuel Alonso-Domínguez, María Magdalena Anguita Arance, Araceli Salamanca Cuenca, Antonio Jiménez-Velasco, Santiago Osorio Prendes, Marta Santaliestra, María José Lis Chulvi, Juan Carlos Hernández-Boluda, Valentín García-Gutiérrez

**Affiliations:** 1Hospital Ramón y Cajal, Instituto Ramón y Cajal de Investigación Sanitaria (IRYCIS), 28034 Madrid, Spain; 2Hospital Duran i Reynals-ICO, 08908 Barcelona, Spain; 3Josep Carreras Leukaemia Research Institute, ICO-Hospital Germans Trias i Pujol, Universitat Autònoma de Barcelona, 08916 Badalona, Spain; 4Hospital Quirón Salud Zaragoza, 50006 Zaragoza, Spain; 5Hospital Virgen de la Arrixaca, 30120 Murcia, Spain; 6Hospital Regional Universitario de Málaga, 29010 Málaga, Spain; 7Hospital Universitario de León, 24071 León, Spain; 8Hospital Universitario y Politécnico La Fe, 46026 Valencia, Spain; 9Hospital General de Alicante, 03010 Alicante, Spain; 10Hospital Universitario de Gran Canaria Doctor Negrín, 35010 Gran Canaria, Spain; 11Hospital La Princesa, 28006 Madrid, Spain; 12Hospital Del Mar, 08003 Barcelona, Spain; 13Hospital Clínico Universitario de Salamanca, 37007 Salamanca, Spain; 14Hospital Virgen Macarena, 41009 Sevilla, Spain; 15Hospital Doctor José Molina Orosa de Lanzarote, 35500 Las Palmas, Spain; 16Hospital Clínic, 08036 Barcelona, Spain; 17Hospital General de Granollers, 08402 Barcelona, Spain; 18Hospital Clínico Universitario de Santiago de Compostela, 15706 A Coruña, Spain; 19Hospital General de Castellón, 12004 Castellón, Spain; 20Hospital Universitario Dr. J Trueta-CO, 17007 Girona, Spain; 21Hospital Universitario Virgen de la Victoria, 29010 Málaga, Spain; 22Hospital Universitario de Canarias, 38320 Tenerife, Spain; 23Hospital Universitario Marqués de Valdecilla, 39008 Santander, Spain; 24Hospital Parc Tauli, 08208 Sabadell, Spain; 25Hospital Virgen de la Luz, 16002 Cuenca, Spain; 26Hospital Universitario de Burgos, 09006 Burgos, Spain; 27Fundación Hospital Alcorcón, 28922 Madrid, Spain; 28Hospital Universitario La Paz, 28046 Madrid, Spain; 29Hospital Universitario Puerto Real, 11510 Cádiz, Spain; 30Hospital Dr. Peset, 46017 Valencia, Spain; 31Hospital Virgen de la Salud, 45005 Toledo, Spain; 32Hospital Universitario Fundación Jiménez Díaz, 28040 Madrid, Spain; 33Complejo Hospitalario de Jaén, 23007 Jaén, Spain; 34Hospital de Jerez de la Frontera, 11407 Cádiz, Spain; 35Hospital General Universitario Gregorio Marañón, 28007 Madrid, Spain; 36Hospital Universitari Mútua Terrassa, 08221 Barcelona, Spain; 37Hospital General Universitario de Valencia, 46014 Valencia, Spain; 38Hospital Clínico Universitario de Valencia, 46010 Valencia, Spain

**Keywords:** asciminib, drug intolerance, toxicities, chronic myeloid leukemia

## Abstract

**Simple Summary:**

After the recent irruption of asciminib into the therapeutic arsenal for chronic myeloid leukemia, real-life data remain scarce to determine which patients may benefit most from this drug. Data on the efficacy of the drug in real-world setting have been reported, but a detailed analysis of the toxicity profile and the influence of prior intolerance to classical tyrosine kinase inhibitors (TKIs) has not been performed. The aim of the present analysis is to study in detail the toxicity profile of asciminib as well as to describe the risk of cross-toxicity with classical TKIs. These results may help to select the patient profile with the best chance of therapeutic success with asciminib monotherapy.

**Abstract:**

(1) Background: Despite the prognostic improvements achieved with tyrosine kinase inhibitors (TKIs) in chronic myeloid leukemia (CML), a minority of patients still fail TKIs. The recent introduction of asciminib may be a promising option in intolerant patients, as it is a first-in-class inhibitor with a more selective mechanism of action different from the ATP-competitive inhibition that occurs with TKIs. Therefore, our goal was to analyze toxicities shown with asciminib as well as to study cross-toxicity with previous TKIs. (2) Methods: An observational, multicenter, retrospective study was performed with data from 77 patients with CML with therapeutic failure to second-generation TKIs who received asciminib through a managed-access program (MAP) (3) Results: With a median follow-up of 13.7 months, 22 patients (28.5%) discontinued treatment: 32% (7/22) due to intolerance and 45% (10/22) due to resistance. Fifty-five percent of the patients reported adverse effects (AEs) with asciminib and eighteen percent grade 3–4. Most frequent AEs were: fatigue (18%), thrombocytopenia (17%), anemia (12%), and arthralgias (12%). None of the patients experienced cardiovascular events or occlusive arterial disease. Further, 26%, 25%, and 9% of patients required dose adjustment, temporary suspension, or definitive discontinuation of treatment, respectively. Toxicities under asciminib seemed lower than with prior TKIs for anemia, cardiovascular events, pleural/pericardial effusion, diarrhea, and edema. Cross-toxicity risk was statistically significant for thrombocytopenia, anemia, neutropenia, fatigue, vomiting, and pancreatitis. (4) Conclusion: Asciminib is a molecule with a good safety profile and with a low rate of AEs. However, despite its new mechanism of action, asciminib presents a risk of cross-toxicity with classical TKIs for some AEs.

## 1. Introduction

The discovery of the pathophysiology of chronic myeloid leukemia (CML) made possible to design drugs focused on a molecular target (BCR::ABL1), opening a new era in anti-tumor therapy through tyrosine kinase inhibitors (TKIs) [1]. Thanks to the advent of imatinib and subsequently to new-generation TKIs, the overall survival of CML patients has matched that of the general population. This fact has meant that the prevalence of CML has markedly increased in recent years [2,3].

All current approved TKIs (imatinib, dasatinib, nilotinib, bosutinib, and ponatinib) have a common mechanism of action by competitive binding to the adenosine triphosphate (ATP) site in the ABL kinase from the BCR::ABL1 fusion protein [4]. The wide homology between the ATP-binding sites of many human kinases leads to a non-specific inhibition of these, which is associated with the development of “off-target” adverse effects (AEs) [2,4,5,6].

This “off target” phenomenon of TKIs may decrease tolerability and in some patients can lead to significant long-term safety issues or even an impact on quality of life over time. Approximately 9% to 25% of patients receiving first-line TKIs discontinue treatment due to adverse reactions [7,8,9,10]. Once therapeutic failure due to intolerance has occurred, the risk of developing AEs that prevent treatment success is much higher with subsequent lines, making these patients therapeutically challenging [11,12].

Asciminib was recently approved in 2021 by the Food and Drug Administration (FDA) and in 2022 by the European Medicines Agency (EMA) for the treatment of CML after failure of two lines of therapy based on the ASCEMBL trial [13]. This drug, unlike currently approved ATP-competitive TKIs, binds to the myristoyl pocket of ABL1, resulting in an allosteric inhibition of the BCR::ABL1 kinase activity [4]. This distinct mechanism of action is much more selective so that side effects related to inhibition of non-BCR::ABL1 kinases have been expected to be greatly diminished [14,15].

A significant proportion of patients fail TKIs therapy due to the development of mutations in the ABL-ATP binding site with successive treatments, which may cause limited sensitivity to the remaining TKIs [16]. Multidrug-resistant patients and those with T315I mutation or with compound mutations currently represent a group with an unmet therapeutic need [4,17,18]. Because asciminib does not bind to the ATP binding site, it is expected to maintain substantial activity against kinase domain mutations that confer acquired drug resistance to ATP-competitive TKIs [5].

In the phase 1 dose-escalation clinical trial in 150 patients, asciminib showed clinical activity and a good safety profile [19]. In the phase 3 clinical trial comparing asciminib versus bosutinib in patients with two or more TKIs, the superiority of the primary endpoint of higher MMR rate at week 24 was achieved and showed an improved safety profile as compared with bosutinib [13].

Therefore, the rate of therapeutic failures due to intolerance and the appearance of mutations and other resistance mechanisms that appear in patients with increasingly longer survival makes it necessary to develop new drugs that allow us to meet the primary objectives in CML: long-term survival, safety, and quality of life in our patients. In this sense, it is very relevant to explore real-life data of asciminib, which may be a relevant therapeutic alternative in patients who fail multiple treatments and have few therapeutic options.

Our group reported preliminary efficacy and safety data from our series [20,21]. Other groups have also reported their experience with asciminib in the real-world setting; however, their results have predominantly focused on efficacy without in-depth studies on the drug’s safety profile [22,23]. The aim of the present study is to carry out a detailed analysis of the toxicity of asciminib in real clinical practice as well as to perform a cross-toxicity analysis with classical TKIs. In addition, we aimed to update the efficacy data with a larger number of patients and follow-up time.

## 2. Materials and Methods

An observational, multicenter, retrospective study was conducted with data from 77 CML patients who experienced treatment failure to two or more TKIs. Patients received asciminib between October 2018 and June 2022 through a managed-access program (MAP) provided by Novartis at 38 centers in Spain. The study was approved by the Spanish Medicines Agency (AEMPS) and the Ethics Committee of the Hospital Universitario Ramón y Cajal (Madrid, Spain), with informed consent obtained from all patients.

Blood samples for BCR::ABL1 analysis were processed in EUTOS-accredited laboratories. Response analysis was performed following the European Leukemia Net 2020 recommendations [24]. Treatment failure to previous TKIs was defined as resistance (according to ELN recommended milestones at different treatment time points) or intolerance (unacceptable toxicity leading to discontinuation of the TKI). Mutational status testing was performed by next-generation sequencing (NGS), when feasible, in patients in whom therapeutic failure due to resistance occurred. Clinical data were collected from the medical chart by the responsible CML expert physician using the REDCap electronic database. In addition to demographic information, the AEs presented with each of the previous lines were collected as well as the degree of the adverse effect and whether it led to the need of dose modification/treatment discontinuation. Treatment AEs were graded according to the National Cancer Institute Common Terminology Criteria for Adverse Events Version 5.0. Data analysis was performed with SPSS Version 25.0. Cross-toxicity was defined as occurrence during treatment with asciminib of the same TKI drug-related AE(s). Cross-intolerance was defined as occurrence during treatment with asciminib of the same TKI drug-related AE(s) that led to asciminib discontinuation. For the analysis of cross-toxicity, the frequency of occurrence of the adverse effect in the group of patients who had already experienced this AE was determined versus the frequency in the group of patients who had not experienced it with previous TKIs. Fisher’s exact test was used for comparison. *p* < 0.05 was used to consider the results statistically significant.

## 3. Results

The characteristics of our series are shown in Table 1. The mean age at the time of data collection was 66 years (range 37–92). Eighty-two percent of patients had received ≥3 TKIs previously (range 1–5 previous lines). Thirty-four percent had previously received ponatinib. The median time under treatment with previous TKI was 6.9 years (13 months median time for the first line, 11 months for second line, 21.3 for third line, 14.4 months for fourth line, and 10.4 for the fifth line). The switch to asciminib was due to intolerance in 64% of patients. Except for one patient who started asciminib in the accelerated phase, the rest were in chronic-phase disease. Twenty-five percent harbored BCR::ABL1 mutations (T315I mutation in 4). Compound mutations were not reported in any patient. Asciminib was started at a dose of 40 mg BID with the exception of patients with T315I mutation, who started at 200 mg BID.

With a median follow-up of 13.7 months, 71% of the patients continued treatment with asciminib (Figure 1). Of the total dropouts, 7/22 (9% of total) were due to intolerance, 10/22 due to resistance, and 5/22 due to death from any cause (pharyngeal neoplasm, hepatic adenocarcinoma, skin ulcer with calciphylaxis, and two unknown causes). Of the seven patients who abandoned treatment due to side effects, six of them were caused by side effects that had presented with previous lines.

### 3.1. Safety

Frequency of AEs is shown in Figure 2. Fifty-five percent of patients experienced some AEs with asciminib: most of them mild (grades 1–2), with 18% being grade 3–4. The most frequent AEs were fatigue (18%), thrombocytopenia (17%), anemia (12%), and arthralgias (12%). The most frequent grade 3–4 AEs were thrombopenia (3.9%) and fatigue (3%). None of the patients with previous cardiovascular events presented a new event. There were no cases of de novo occlusive peripheral arterial disease (PAD) although one patient experienced worsening of preexisting PAD. Pancreatitis was observed in two patients, both with the standard dose of 40 mg BID. Other AEs reported in isolation were one grade 3 pneumonitis, one grade 3 creatine phosphokinase (CPK) elevation, one patient with grade 3 renal impairment, one patient with grade 3 muscle spasm, one episode of grade 2 mucositis, two patients with mild skin dryness/scaling, and one patient who presented with grade 1 hypoglycemia.

Comparing the frequency of AEs between resistant and intolerant patients, we observed that there are no significant differences (Figure 3).

In 20 patients (26%) a dose adjustment to 20 mg twice daily was necessary due to intolerance. Temporary discontinuation of treatment was needed in 19 patients (25%). The need for dose adjustment was more frequent in the intolerant group than in the group resistant to previous lines (33% vs. 14%). Patients who required a dose reduction had an MMR rate at the end of follow-up of 55% (11/20) vs. 63% (34/54) in those who maintained standard doses. Nine percent (7/77) of patients had to stop treatment definitively due to side effects (pleural effusion, pneumonitis, renal failure, worsening of PAD, thrombocytopenia, and two due to pancreatitis). Of the patients who discontinued due to side effects, 86% (6/7) received asciminib due to intolerance to previous lines. Six of the seven patients intolerant to asciminib failed this drug for cross-intolerance (an adverse effect that had already led to discontinuation a previous TKI).

A comparison of AEs of asciminib versus previous TKIs is shown in Figure 4. Regarding cytopenias (Figure 4a), a mean of 16.9% of patients presented thrombocytopenia with asciminib vs. 23.7% with previous lines. For anemia and neutropenia, the percentages were 11.7% vs. 22.9% and 6.5% vs. 10.5%.

For non-hematologic toxicities (Figure 4b), the difference in frequency of AEs seemed to favor of asciminib for the percentage of cardiovascular events (0% vs. 5.6%), pleural/pericardial effusion (5.2% vs. 13.9%), diarrhea (1.3% vs. 13.9%), and edema (2.7% vs. 13.9%). The differences observed for the rate of arthralgias (11.7% vs. 10.5%), headache (1.3% vs. 7%), loss of appetite (2.6% vs. 7%), abdominal pain (5.2% vs. 8.0%), fatigue (18.2% vs. 14.3%), nausea (7.8% vs. 11.8%), vomiting (2.6% vs. 6.3%), pancreatitis (2.6% vs. 2%), and rash (5.2% vs. 8.0%) were less pronounced.

### 3.2. Cross-Toxicity and Cross Intolerance

Cross-toxicity with previous TKIs was analyzed for the most frequent AEs observed with asciminib (Table 2). This risk was statistically significant for thrombocytopenia (43% vs. 2%), anemia (22% vs. 4.6%), neutropenia (21% vs. 1.7%), fatigue (35% vs. 12%), vomiting (16.6% vs. 0%), and pancreatitis (33% vs. 0%). Cross-toxicity does not appear to affect the occurrence of cardiovascular events, edema, abdominal pain, diarrhea, or rash. Cross-intolerance led to treatment discontinuation in six patients (pleural effusion, pneumonitis, worsening of PAD, thrombocytopenia, and two cases of pancreatitis). This implies that of the total number of treatment failure due to intolerance, 86% (6/7) did so because of side effects that had already led to treatment failure of a previous TKI.

For patients previously treated with ponatinib, there was no increased frequency of adverse effects or increased risk of cross-toxicity compared to those not previously exposed to ponatinib (Appendix A).

### 3.3. Efficacy

In terms of efficacy (Table 3), 73% of patients maintained or achieved a complete cytogenetic response (CCyR) and 60% a major molecular response (MMR) after almost fourteen months of follow up. Of the patients who had no prior response, 50% achieved CCyR, and 49% achieved MMR. Responses were better in patients who started asciminib for intolerance versus those who started it for resistance (80% of intolerant patients maintained or achieved MMR vs. 26% in the resistant group).

## 4. Discussion

The therapeutic success in CML and the consequent increase in life expectancy means that the prevalence of patients with therapeutic failure to TKIs is also growing every day. As mentioned, once a patient presents a therapeutic failure to the first line, the chances of failure to subsequent lines are much higher (up to 60–70% in second/third lines) [16]. This can be partly explained by the homogeneity of the therapeutic arsenal available to date, in which the mechanism of action of all these drugs is based on ATP competitive inhibition [25].

In this manuscript, we have analyzed, to our knowledge, the largest series of patients treated with asciminib in the real-world setting, aiming to shed some light on several unanswered questions, such as whether asciminib would change the toxicity profile of classical TKIs or overcome cross-intolerance with previous lines.

Overall, the toxicity profile of asciminib has been shown to be favorable in our series, with no unexpected class-specific adverse effects observed. The type of AEs noted, despite its novel mechanism of action, is similar to that of other TKIs.

With respect to that described in clinical trials, a lower percentage of cytopenias and extra-hematological adverse effects were reported compared to the phase 1 dose-escalation clinical trial [19]. The fact that in the dose-escalation trial almost half of the patients received doses greater than 40 mg BID probably contributes to this finding as well as the retrospective nature of the study. With respect to the phase 3 clinical trial ASCEMBL comparing asciminib vs. bosutinib [13], where all the patients received 40 mg BID, it is worth noting the lower rate of thrombocytopenia (17% vs. 29%, with thrombocytopenia grade ≥3, 4% vs. 21%) and neutropenia (6.5% vs. 21%, with neutropenia grade ≥3, 2.6% vs. 17%) in our study, with similar rates of anemia (11.7% vs. 9.6%, grade 3 0% vs. 1.3%). This could be due to differences in the profile of patients between the clinical trial and our study, with slightly longer follow-up in the clinical trial (13.7 vs. 14.9 months) and underreporting in our retrospective study.

In our study, pancreatitis was seen in two patients (2.6%), both of whom had a previous event of pancreatitis grade 3–4 that led to discontinuation of a previous line. In the phase 1 trial, clinical pancreatitis occurred in five patients, all of whom received doses greater than 40 mg BID (three patients receiving 80 mg twice daily, one receiving 150 mg twice daily, and one receiving 200 mg once daily). Of interest, three of the five patients had had pancreatitis when using a previous TKI. Asymptomatic biochemical elevations of lipase or amylase level occurred in 35 additional patients at all doses except 10 mg twice daily. These events were self-limited and did not progress to clinical pancreatitis. In the ASCEMBL trial, pancreatitis was not observed in any patient; however, it is important to note that patients with previous pancreatitis in the last 12 months prior to treatment were excluded.

The phase 3 trial comparing to bosutinib showed a 3.2% percentage of patients treated with asciminib suffering from arterial occlusive events (AOEs). Since this percentage was higher than that observed with bosutinib (1.3%), the need to closely monitor the possible relationship of asciminib and AOEs has been postulated. Of interest, we found no new AOEs in our heavily pretreated patients (only one case of worsening of previous PAD), decreasing to some extent the concerns about this particularly important risk at least at the short term.

Some grade 1–2 adverse effects were reported less frequently than in the phase 3 trial (headache, diarrhea, back pain, nasopharyngitis, upper respiratory tract infections, etc.), probably related to the retrospective nature of the study. Of note, the percentage of hypertension ≥3 was 1.3% in our study vs. 5.8% registered in ASCEMBL.

When compared with classical TKIs, the frequency of cytopenias, cardiovascular events, pleural effusion, diarrhea, and edema seemed to be lower, so asciminib may be an alternative to consider in patients at risk of these toxicities. It should be noted that the median follow-up time for each of the previous lines was somewhat different, and this may affect the frequency of adverse effects reported with each of them (13.7 months for asciminib, 13 months for the first line, 11 months for second line, 21.3 for third line, 14.4 moths for fourth line, and 10.4 for the fifth line).

Analyzing cross-toxicity is very relevant due to the implication it usually has in the successive therapeutic failures in intolerant patients and in the decision of a therapeutic alternative. In this regard, asciminib seems to maintain the risk of cross-toxicity for some of the AEs (cytopenias, fatigue, vomiting, and pancreatitis). Nevertheless, cross-toxicity does not seem to affect the occurrence of cardiovascular events, edema, abdominal pain, diarrhea, or rash, which makes it a good alternative in patients who discontinued classical TKIs for these reasons.

Likewise, despite the risk of cross-toxicity for the AEs mentioned above, asciminib may still be an acceptable option given that the rates in the population previously affected by such toxicity are not very high for anemia (22% among the population with previous anemia), neutropenia (21% among the population with previous neutropenia). and vomiting (16.6% among the population with previous vomiting). However, in cases of previous severe thrombocytopenia or pancreatitis, the risk of cross-intolerance remains high (43% and 33%, respectively), so the advisability of initiating this drug in patients who discontinue due to these EAs should be assessed with caution.

Six out of the seven patients that discontinued treatment in our series due to intolerance did so because of AEs that had previously led to discontinuation of a previous TKIs (pleural effusion, pneumonitis, worsening of PAD, thrombocytopenia, and two cases of pancreatitis), which highlights the weight of therapeutic failure due to crossed-toxicity (86% of all dropouts due to intolerance). A larger number of patients and follow-up time verify preliminary data previously reported by our group, with similar rates of adverse effects and a potential risk of cross-intolerance for some toxicities [20,21].

In terms of efficacy, asciminib has been shown to be effective in this heavily treated population, with more than 80% of patients having received three or more lines of treatment. It shows excellent results in intolerant patients, with MMR rates >80%, with much more modest results in the resistant patient population (MMR 26%).

The response rates in the resistant population (CCyR 46% and MMR 26%) show that these patients remain a therapeutic challenge. Therefore, in addition to ATP binding site mutations, other mechanisms previously described as potential contributors to resistance may play a role, such as an increased activation of signaling pathways, activation of drug transporters, epigenetic dysregulation, or microenvironmental factors [26].

Therapeutic alternatives in this population are scarce beyond clinical trials or allogeneic hematopoietic stem cell transplantation, which, as is known, has high morbidity and mortality. In these patients, the results of ongoing studies that propose combining classical TKIs with asciminib will be of great interest [27,28,29,30]. The presence of an alternative inhibitory target offers the possibility of simultaneously using myristoyl and ATP binding sites as targets to enhance kinase inhibition. This mechanism of action could have potential benefit in multidrug-resistant patients or patients with blast-phase CML [2,17].

Overall, the odds of treatment discontinuation due to AEs was low, with only 9% of patients failing the treatment due to intolerance after a median follow-up of 13.7 months. This low percentage of treatment discontinuations due to AEs is more relevant considering that more than 60% of patients received asciminib due to intolerance to prior TKIs. If we consider exclusively the intolerant population, discontinuations were 12.2% (6/49) in this group. These data are in line with the results of phase 1 and phase 3 trials, where the proportion of patients who experienced AEs leading to treatment discontinuation was less than 10%.

It is important to highlight the limitations of our study, which firstly include the challenge of collecting data retrospectively, with certainty higher rate of unreported adverse effects; the possible lack of homogeneity in data collection given the multicenter nature; and the short follow-up time. The strengths include the number of patients recruited due to its multicenter nature and the expertise of the physicians in charge of data collection.

## 5. Conclusions

Asciminib may be an adequate alternative to achieve our goals of survival, safety, and quality of life in CML patients, particularly in those with intolerance to previous lines. Our data show a low frequency of adverse effects for relevant toxicities such as cardiovascular events or pleural/pericardial effusion with this drug. The risk of cross-toxicity is maintained for some adverse effects such as pancreatitis or thrombocytopenia. Other relevant toxicities (pleural/pericardial effusion, edema, or diarrhea) do not seem to be affected by this phenomenon, which increases the probability of therapeutic success in patients who have repeatedly failed successive TKIs for these reasons. In the group of resistant patients, the results are modest, and it remains to be seen whether the possibility of combinations with classical TKIs can improve outcomes in a patient group that remains a challenge today.

## Figures and Tables

**Figure 1 cancers-15-01045-f001:**
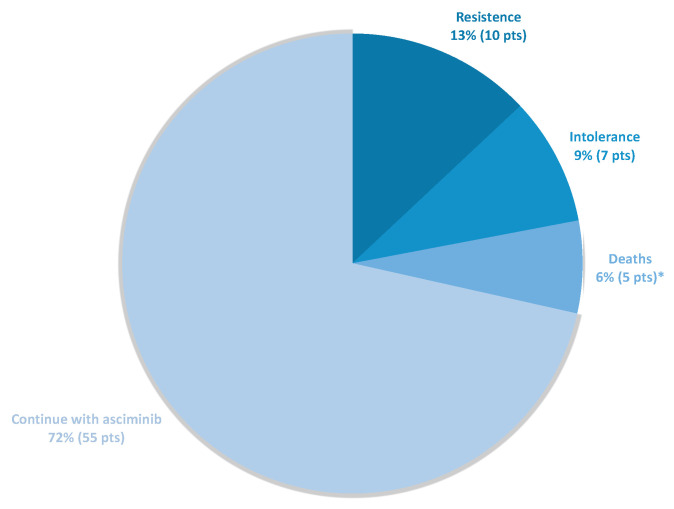
Discontinuations of asciminib divided by cause at the end of the follow-up period. Pts, patients. * Causes of deaths: pharyngeal neoplasm, hepatic adenocarcinoma, skin ulcer with calciphylaxis, and two unknown causes.

**Figure 2 cancers-15-01045-f002:**
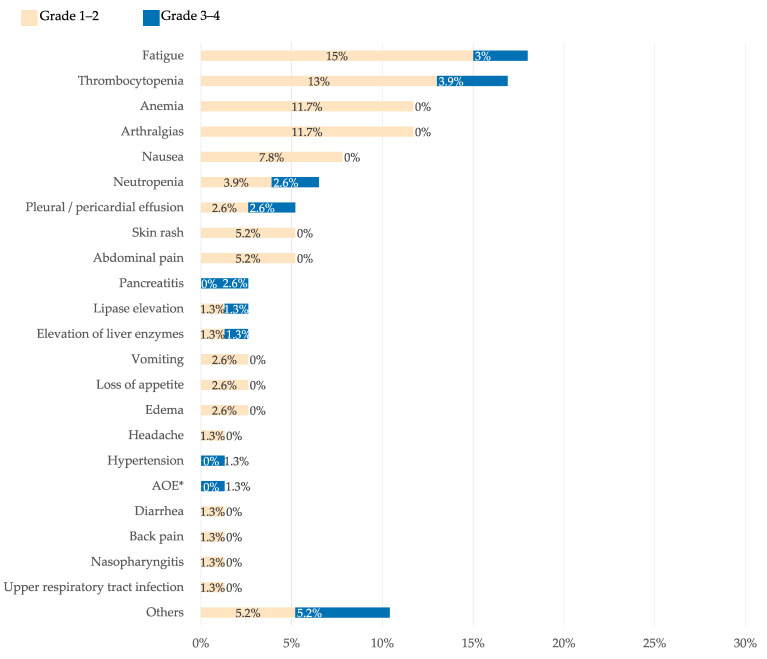
Frequency of adverse effects observed with asciminib. In pale orange: frequency of grade 1–2 adverse effects. In blue color: frequency of grade 3–4 adverse effects. AOE, arterial occlusive event. * Worsening of previous peripheral arterial disease.

**Figure 3 cancers-15-01045-f003:**
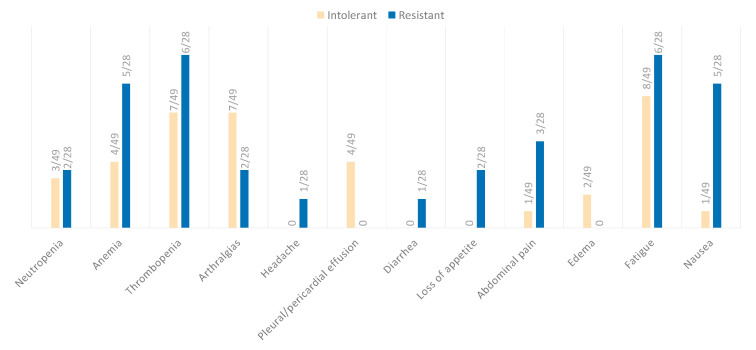
Frequency of adverse effects with asciminib in intolerant (pale orange) versus resistant (blue) patients.

**Figure 4 cancers-15-01045-f004:**
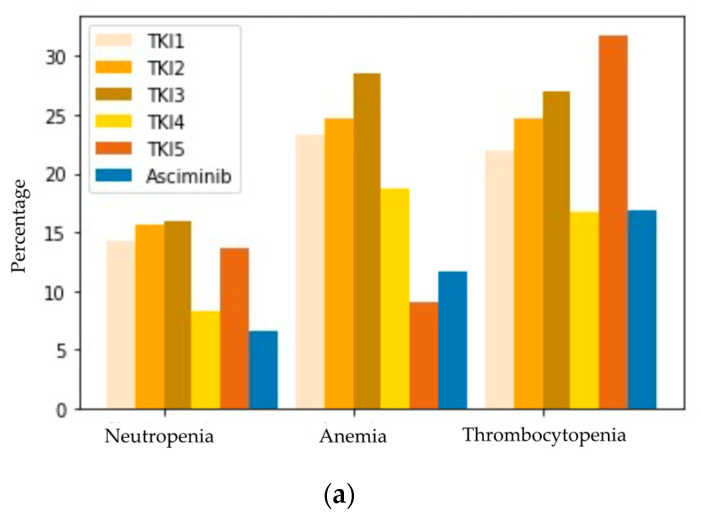
(**a**) Frequency of hematologic toxicity with previous TKI versus asciminib. (**b**) Frequency of extra-hematologic toxicity with previous TKI versus asciminib. In pale orange to brown: the reported frequency of cytopenias for the different lines prior to asciminib. TKI1 refers to the first line of treatment and TKI2–5 to the successive lines. All previous lines correspond to one of the classical TKIs: either imatinib, dasatinib, nilotinib, bosutinib, or ponatinib. In blue color: the frequency reported with asciminib.

**Table 1 cancers-15-01045-t001:** Baseline characteristics of the cohort.

	Patients (*n* = 77)
Median age at data collection, y (range)	66 (37–92)
Median age at diagnosis, y (range)	52 (20–87)
Female sex, *n* (%)	38 (49)
Median time on previous TKIs, y (range)	6.9 (0.5–29)
Disease stage before asciminib, *n* (%)	
Chronic phase	76 (99)
Accelerated phase	1 (1)
Blast phase	0
Sokal risk, *n* (%)	
Low	35 (45)
Intermediate	19 (24)
High	13 (17)
Unknown	10 (12)
Switch to asciminib due to intolerance, *n* (%)	49 (64)
Switch to asciminib due to resistance, *n* (%)	28 (36)
TKI at diagnosis, *n* (%)	
Imatinib	59 (77)
Dasatinib	8 (10)
Nilotinib	9 (12)
Bosutinib	1 (1)
≥3 prior TKI lines, *n* (%)	63 (82)
Prior use of ponatinib, *n* (%)	26 (34)
BCR::ABL1 mutations, *n* (%)	19 (25)
T315I, *n* (%)	4 (5)

y, years; TKI, tyrosine kinase inhibitors.

**Table 2 cancers-15-01045-t002:** Frequency of different toxicities with each of the lines of treatment and with asciminib. At the bottom: cross-toxicity analysis. Frequency in the group with a history of the adverse effect and frequency in the group without a history of the adverse effect. AE, adverse effect.

	Thombocytopenia	Anemia	Neutropenia	Cardio-Vascular Events	Arthralgias	Headache	Pleural/Pericardial Effusion	Diarrhea	Loss of Appetite	Abdominal Pain	Edema	Fatigue	Nausea	Vomiting	Pancreatitis	Skin Rash
ITK1	17/77 (22%)	18/77 (23.4%)	11/77 (14.3%)	1/77 (1.3%)	13/77 (16.8%)	5/77 (6.4%)	3/77 (3.8%)	11/77 (14.2%)	7/77 (9.1%)	7/77 (9.1%)	18/77 (23.4%)	10/77 (13%)	13/77 (16.8%)	7/77 (9.1%)	1/77 (1.3%)	11/77 (14.3%)
ITK2	19/77 (24.7%)	19/77 (24.7%)	12/77 (15.6%)	4/77 (5.1%)	5/77 (6.5%)	6/77 (7.8%)	20/77 (26%)	6/77 (7.8%)	5/77 (6.5%)	6/77 (7.8%)	10/77 (13%)	15/77 (19.5%)	6/77 (7.8%)	4/77 (5.2%)	2/77 (2.6%)	5/77 (6.5%)
ITK3	17/63 (27%)	18/63 (28.6%)	10/63 (15.9%)	9/63 (14%)	6/63 (9.5%)	5/63 (7.9%)	12/63 (19%)	12/63 (19%)	3/63 (4.7%)	4/63 (6.3%)	4/63 (6.3%)	9/63 (14.3%)	5/63 (7.9%)	0/63 (0%)	2/63 (3.2%)	4/63 (6.3%)
ITK4	8/48 (16.7%)	9/48 (18.7%)	4/48 (8.3%)	0/48 (0%)	1/48 (2.1%)	3/48 (6.2%)	4/48 (8.3%)	15/48 (31.2%)	3/48 (6.2%)	4/48 (8.3%)	5/48 (10.4%)	4/48 (8.3%)	8/48 (16.7%)	5/48 (10.4%)	1/48 (2.1%)	0/48 (0%)
ITK5	7/22 (31.8%)	2/22 (9.1%)	3/22 (13.6%)	2/22 (9%)	5/22 (22.7%)	1/22 (4.5%)	1/22 (4.5%)	3/22 (13.6%)	2/22 (9.1%)	2/22 (9.1%)	3/22 (13.6%)	3/22 (13.6%)	2/22 (9.1%)	2/22 (9.1%)	0/22 (0%)	3/22 (13.6%)
Asciminib	13/77 (16.9%)	9/77 (11.7%)	5/77 (6.5%)	0 (0%)	9/77 (11.7%)	1/77 (1.3%)	4/77 (5.2%)	1/77 (1.3%)	2/77 (2.6%)	4/77 (5.2%)	2/77 (2.6%)	14/77 (18.2%)	6/77 (7.8%)	2/77 (2.6%)	2/77 (2.6%)	4/77 (5.2%)
Risk of developing toxicity with asciminib in patients WITH that prior EEAA.	12/28 (43%)	7/32 (22%)	4/19 (21%)	0/15 (0%)	5/22 (23%)	0/14 (0%)	3/29 (10%)	1/38 (2.6%)	1/12 (8%)	1/17 (5.8%)	1/26 (3.8%)	7/20 (35%)	3/22 (13.6%)	2/12 (16.6%)	2/6 (33%)	1/17 (5.8%)
Risk of developing toxicity with asciminib in patients WITHOUT that prior EEAA.	1/49 (2%)	2/44 (4.6%)	1/58 (1.7%)	0/62 (0%)	4/55 (7%)	1/63 (1.6%)	1/48 (2%)	0/39 (0%)	1/65 (1.5%)	3/60 (5%)	1/51 (2%)	7/57 (12%)	3/55 (5.4%)	0/65 (0%)	0/71 (0%)	3/60 (5%)
*p*-value	0.000	0.029	0.012	1.000	0.109	1.000	0.147	0.494	0.289	1.000	1.000	0.040	0.345	0.023	0.005	1.000

**Table 3 cancers-15-01045-t003:** Grades of overall responses to asciminib taking into account intolerance vs. resistance. CHR, complete hematological response; CCR, complete cytogenetic response; MMR, major molecular response; MR4.5, detectable disease with BCR::ABL1IS < 0.0032%. ^a^ Probability of achieving or maintaining previous response. ^b^ Patients without response at baseline.

Response to Asciminib	Resistant (*n* = 27)	Intolerant (*n* = 47)	Total (*n* = 74)
CCR ^a^	12/27 (44%)	42/47 (89%)	54/74 (73%)
MMR ^a^	7/27 (26%)	38/47 (80%)	45/74 (60%)
MR4.5 ^a^	4/27 (15%)	19/47 (40%)	23/74 (31%)
Patients without response at baseline			
CCR ^b^	5/20 (25%)	13/16 (81%)	18/36 (50%)
MMR ^b^	5/25 (20%)	22/30 (73%)	27/55 (49%)
MR4.5 ^b^	4/27 (15%)	16/44 (32%)	20/71 (28%)

## Data Availability

The data presented in this study are available on request from the corresponding author. The data are not publicly available due to privacy of participating patients and hospitals.

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
