# Peer review of "Toxicity of Asciminib in Real Clinical Practice: Analysis of Side Effects and Cross-Toxicity with Tyrosine Kinase Inhibitors"

_cancers, 2023, doi:10.3390/cancers15041045_

Round 1

Reviewer 1 Report

Very interesting paper about the real world AEs and cross intolerance on asciminib,  a new class of bcr-abl inhibitor. The paper summarize the Spanish experience group on this drug regarding side effects as well as cross intolerance with previous approved TKIs. The paper is relevant for the clinical practice as describe this experience outside from a more control clinical trial and therefore reflects more accurate how the drug perform in real life in heavily pre treated population of CP-CML. Overall the paper is ready to be accepted but may benefit of some English editing mainly in the discussion. as well as to change to colors on the figures to be more easily identified. 

Author Response

Dear Reviewer,

Thank you very much for your valuable suggestions. These are our reactions to your proposed changes:

  • English editing mainly in the discussion. We have revised the English again, applying changes that you can see throughout the manuscript.
  • Change colors on the figures to be more easily identified.We have modified the colors in Figure 4 to make the columns more easily distinguishable.

Thank you again for taking the time to carefully review our article. We look forward to hearing from you if you consider further modifications to the manuscript.

Best regards.

Reviewer 2 Report

Having another effective TKI in the treatment arsenal is a great development for patients with CML. Here, Authors collected asciminib treatment related AEs. 55 of 77 pts experienced any AE with asciminib; 7 stopped due to AEs (pleural effusion, pneumonitis, renal failure, worsening of PAD).

Authors should elaborate on how they captured the data retrospectively; interviewed patients? Reviewed medical charts? In methods section

Authors should include a paragraph explaining study limitations – challenging nature of collecting AE data retrospectively, possibility of missing some of the AEs, short follow etc….

Authors compare asciminib AEs with other TKI AEs. Median follow up of asciminib was 14 months. Authors need to report the median follow-up with other TKI, as duration of follow up will affect the cumulative incidence of AEs. For example, it is not fair to state (line 229) that the frequency of AEs was statistically significant in favor of asciminib for the percentage of cardiovascular events (0% vs 5,6%, p=0,03) if median follow up with other TKIs are much longer than the asciminib group. It is more likely to see higher incidence of certain AEs with longer follow-up.

Line 64: Background: Despite the prognostic improvements achieved with tyrosine kinase inhibitors (TKIs) in chronic myeloid leukemia (CML), many patients still fail TKIs ---- this is an overstatement, only a minority of patients fail TKIs

Figure 2 – hypertension misspelled

Figure 4a – very difficult to discern bars, pls choose more distinctive colors

Line 248 -249 : Cross-intolerance risk was statistically significant for thrombocytopenia (43% vs 2%), anemia (22% vs 4.6%), neutropenia (21% vs 1.7%), fatigue (35% vs 12%), vomiting (16.6% vs 0%) and pancreatitis (33% vs 0%). –  state clearly what do you mean by “cross intolerance” or “cross toxicity”

Table 2 – TKI misspelled, thrombocytopenia misspelled

Line 297-307 : already stated in introduction

Line 314 – which phase 3 clinical trial? Reported percentages comparing ? % vs % ?

Line 331 - However, the number of AOEs was superior with asciminib compared to bosutinib – please rephrase this sentence for clarity

Line 333-335 – I disagree; this study cannot back up this claim due retrospective nature and short follow up   

Line 396 - Conclusion – “The risk of cross intolerance, although clearly maintained for some adverse effects such as pancreatitis or thrombocytopenia, disappears for other relevant toxicities, which increases the probability of therapeutic success in patients who have repeatedly failed successive TKIs” – this is a long and confusing and difficult to understand paragraph, please simplify

Author Response

Dear Reviewer,

Thank you very much for your valuable suggestions. These are our reactions to your proposed changes:

  • Authors should elaborate on how they captured the data retrospectively; interviewed patients? Reviewed medical charts? In methods section. We have included information about how the information was collected, which was through the patients' medical chart (lines 170-172).
  • Authors should include a paragraph explaining study limitations – challenging nature of collecting AE data retrospectively, possibility of missing some of the AEs, short follow etc…. we have incorporated a paragraph including our study limitations (lines 521-526).
  • Authors compare asciminib AEs with other TKI AEs. Median follow up of asciminib was 14 months. Authors need to report the median follow-up with other TKI, as duration of follow up will affect the cumulative incidence of AEs. For example, it is not fair to state (line 229) that the frequency of AEs was statistically significant in favor of asciminib for the percentage of cardiovascular events (0% vs 5,6%, p=0,03) if median follow up with other TKIs are much longer than the asciminib group. It is more likely to see higher incidence of certain AEs with longer follow-up. This is a very good point. We have added the information about the median time of each of the lines. While it is true that it is not very different, you are right to point out that it is difficult to make a statistical comparison, so we have removed the statistical analysis and left it as a descriptive analysis. We have incorporated this sentence to state that the time of follow up could affect frequencies of adverse effects: “It should be noted that the median follow-up time for each of the previous lines was somewhat different, and this may affect the frequency of adverse effects reported with each of them (13.7 months for asciminib, 13 months for the first line, 11 months for second line, 21.3 for third line, 14.4 moths for fourth line, and 10.4 for the fifth line)”.
  • Line 64: Background: Despite the prognostic improvements achieved with tyrosine kinase inhibitors (TKIs) in chronic myeloid leukemia (CML), many patients still fail TKIs ---- this is an overstatement, only a minority of patients fail TKIs. We have changed the sentence to: “Despite the prognostic improvements achieved with tyrosine kinase inhibitors (TKIs) in chronic myeloid leukemia (CML), a significant proportion of patients still fail TKIs”.
  • Figure 2 – hypertension misspelled. The word was corrected.
  • Figure 4a – very difficult to discern bars, pls choose more distinctive colors. We have modified the colors in Figure 4 to make the columns more easily distinguishable.
  • Line 248 -249: Cross-intolerance risk was statistically significant for thrombocytopenia (43% vs 2%), anemia (22% vs 4.6%), neutropenia (21% vs 1.7%), fatigue (35% vs 12%), vomiting (16.6% vs 0%) and pancreatitis (33% vs 0%). –  state clearly what do you mean by “cross intolerance” or “cross toxicity”. We were not properly defining the term cross-intolerance. We have added the definitions and differentiated the terms cross-toxicity and cross-intolerance as follows:
    • Cross-toxicity was defined as occurrence during treatment with asciminib of the same TKI drug-related AE(s).
    • Cross-intolerance was defined as occurrence during treatment with asciminib of the same TKI drug-related AE(s) that led to asciminib discontinuation.

The analysis that has been mainly performed is that of cross-toxicity. As there were only 7 patients who fail treatment due to intolerance, an extensive analysis of cross-intolerance is not possible due to the sample size. We have changed the title to "Toxicity of asciminib in real clinical practice; Analysis of side effects and cross-toxicity with tyrosine kinase inhibitors", to fit the definition of what we have analyzed.

  • Table 2 – TKI misspelled, thrombocytopenia misspelled. Misspelled words in the table have been corrected.
  • Line 297-307 : already stated in introduction. We have greatly simplified the paragraph, leaving it in 4 lines (previously 10), so as not to repeat information.
  • Line 314 – which phase 3 clinical trial? Reported percentages comparing ? % vs % ?. Here we are referring to the phase 3 clinical trial ASCEMBL, which compares asciminib with bosutinib. This paragraph discusses the comparison of our data with the adverse effect rates of asciminib described in this clinical trial. The reported percentage of adverse effects is clearly lower in our study. Here we argue what may be the possible causes explaining these differences (retrospective nature, shorter follow-up time in our study, etc). The percentages compare the frequency in our study vs. the frequency described in the phase 3 clinical trial with asciminib. We have modified the paragraph for clarification.
  • Line 331 - However, the number of AOEs was superior with asciminib compared to bosutinib – please rephrase this sentence for clarity. In this paragraph we are again talking about the ASCEMBL trial that compared asciminib with Bosutinib. In that trial the percentage of arterial occlusive events (AOE) was superior with asciminib (3.2% vs 1.3%). There was a typo in the text, there were 5 patients with AOE with asciminib, which is 3.2%. Here we discuss the concern in this matter in the medical community. We have rewritten the sentence for clarification.
  • Line 333-335 – I disagree; this study cannot back up this claim due retrospective nature and short follow up. We have changed the statement, remarking that in the short term it has shown to be safe in terms of AOEs in our series.
  • Line 396 - Conclusion – “The risk of cross intolerance, although clearly maintained for some adverse effects such as pancreatitis or thrombocytopenia, disappears for other relevant toxicities, which increases the probability of therapeutic success in patients who have repeatedly failed successive TKIs” – this is a long and confusing and difficult to understand paragraph, please simplify. This sentence was modified and substituted for: “The risk of cross-toxicity is maintained for some adverse effects such as pancreatitis or thrombocytopenia. Other relevant toxicities (pleural/pericardial effusion, edema or diarrhea) do not seem to be affected by this phenomenon, which increases the probability of therapeutic success in patients who have repeatedly failed successive TKIs for these reasons.”

Thank you again for taking the time to carefully review our article, your suggestions have greatly improved the manuscript. We look forward to hearing from you if you consider further modifications to the manuscript.

Best regards.

Round 2

Reviewer 2 Report

Line 64: "Despite the prognostic improvements achieved with tyrosine kinase inhibitors (TKIs) in chronic myeloid leukemia (CML), a significant proportion of patients still fail TKIs." 

I guess we are arguing semantics; most frontline CML patients I treat do not fail TKIs. 

I can understand why authors would like to include such sentence, but I recommend them them to edit it. Because a small proportion of patients (5-15%) fail TKIs. Yes. although failures are not very common, it is very important to find better drugs. 

Author Response

Dear Reviewer,

Thank you very much for the early response and taking the time one more time to review our corrections. These are the changes to your suggestions:

  • Line 64: "Despite the prognostic improvements achieved with tyrosine kinase inhibitors (TKIs) in chronic myeloid leukemia (CML), a significant proportion of patients still fail TKIs."  I guess we are arguing semantics; most frontline CML patients I treat do not fail TKIs. I can understand why authors would like to include such sentence, but I recommend them them to edit it. Because a small proportion of patients (5-15%) fail TKIs. Yes. although failures are not very common, it is very important to find better drugs. We have modified the sentence you point out as follows: “Despite the prognostic improvements achieved with tyrosine kinase inhibitors (TKIs) in chronic myeloid leukemia (CML), a minority of patients still fail TKIs”.

Thank you again for your revision, we look forward to hearing from you if you have any additional suggestions.

Best regards.